# Hippocampal place cells construct reward related sequences through unexplored space

**H Freyja Ólafsdóttir[1,2]\*[†], Caswell Barry[2][†], Aman B Saleem[3], Demis Hassabis[4][‡], Hugo J Spiers[1]\*[‡]**

[1]Institute of Behavioural Neuroscience, Department of Experimental Psychology, Division of Psychology and Language Sciences, University College London, London, United Kingdom; [2]Department of Cell and Developmental Biology, University College London, London, United Kingdom; [3]UCL Institute of Ophthalmology, University College London, London, United Kingdom; [4]Gatsby Computational Neuroscience Unit, University College London, London, United Kingdom

**Abstract** Dominant theories of hippocampal function propose that place cell representations are formed during an animal's first encounter with a novel environment and are subsequently replayed during off-line states to support consolidation and future behaviour. Here we report that viewing the delivery of food to an unvisited portion of an environment leads to off-line pre-activation of place cells sequences corresponding to that space. Such '*preplay*' was not observed for an unrewarded but otherwise similar portion of the environment. These results suggest that a hippocampal representation of a visible, yet unexplored environment can be formed if the environment is of motivational relevance to the animal. We hypothesise such goal-biased preplay may support preparation for future experiences in novel environments.

**\*For correspondence:**
h.olafsdottir@ucl.ac.uk (HFÓ);
h.spiers@ucl.ac.uk (HJS)

[†]These authors contributed equally to this work

[‡]These authors are co-senior authors

**Competing interests:** The authors declare that no competing interests exist.

**Reviewing editor**: Howard Eichenbaum, Boston University, United States

## Introduction

We investigated whether the presence of an inaccessible goal in an unvisited portion of an environment was sufficient to elicit pre-activation ('preplay') of hippocampal place cell sequences that will subsequently represent runs through the unvisited environment. To this end, we recorded from ensembles of place cells (*O'Keefe and Dostrovsky, 1971*) (4 rats, 37–66 place cells each, 212 cells in total) while rats ran along a T-shaped track (*Figure 1—figure supplement 1*, *Table 1*) with visible yet inaccessible arms (*Figure 1A*)—RUN1. One arm (counter balanced between animals) was subsequently cued with food while the animal remained on the track—GOAL-CUE. During a rest period before RUN1 (REST1) and after GOAL-CUE (REST2), spiking events—periods of 300 ms or less, where at least 15% of cells were active (*Foster and Wilson, 2006*; *Diba and Buzsaki, 2007*)—were analysed. These spiking events were associated with significantly higher power in the ripple spectrum (80–250 Hz) than other comparable periods (*Figure 1—figure supplement 2*). To investigate whether paths on the cued and uncued arms were preplayed we assessed the match between the order in which cells fired during spiking events and during future runs on the arms (RUN2, *Figure 1—figure supplement 3*). Specifically, we computed the rank-order correlations between spiking events and sequences of place cells active on the arms, referred to as templates (*Lee and Wilson, 2002*; *Foster and Wilson, 2006*; *Diba and Buzsaki, 2007*; *Dragoi and Tonegawa, 2011*) (*Figure 1—figure supplement 4*). Preplay events were identified as those with either a significant positive or negative correlation—a two-tailed test, each tail tested at the 97.5% level. These preplay events were found to exhibit higher power in the ripple spectrum than non-significant spiking events (*Figure 1—figure supplement 2*).

**eLife digest** As an animal explores an area, part of the brain called the hippocampus creates a mental map of the space. When the animal is in one location, a few neurons called 'place cells' will fire. If the animal moves to a new spot, other place cells fire instead. Each time the animal returns to that spot, the same place cells will fire. Thus, as the animal moves, a place-specific pattern of firing emerges that scientists can view by recording the cells' activity and which can be used to reconstruct the animal's position.

After exploring a space, the hippocampus may replay the new place-specific pattern of activity during sleep. By doing so, the brain consolidates the memory of the space for return visits. Recent evidence now suggests that these mental rehearsals—or internal simulations of the space—may begin even before a new space has been explored.

Now, Ólafsdóttir, Barry et al. report that whether an animal's brain simulates a first visit to a new space depends on whether the animal anticipates a reward. In the experiments, rats were allowed to run up to the junction in a T-shaped track. The animals could see into each of the arms, but not enter them. Food was then placed in one of the inaccessible arms. Ólafsdóttir, Barry et al. recorded the firing of place cells in the brain of the animals when they were on the track and during a rest period afterwards. The rats were then allowed onto the inaccessible arms, and again their brain activity was recorded.

In the rest period after the rats first viewed the inaccessible arms, the place cell pattern that would later form the mental map of a journey to and from the food-containing arm was pre-activated. However, the place cell pattern that would become the mental map of the other inaccessible arm was not activated before the rat explored that area. Therefore, Ólafsdóttir, Barry et al. suggest that the perception of reward influences which place cell pattern is simulated during rest. An implication of these findings is that the brain preferentially simulates past or future experiences that are deemed to be functionally significant, such as those associated with reward. A future challenge will be to determine whether this goal-related simulation of unvisited spaces predicts and is needed for behaviour such as successful navigation to a goal.

To establish significance at the population level, the proportion of preplay events measured was compared to a null distribution generated by calculating correlations between place cell templates and shuffled sequences from events (see *Figure 1B–C*).

## Results

During GOAL-CUE, all four animals displayed more interest in the cued arm than the uncued arm (as indexed by the difference in time spent on the cued side of the stem vs the uncued side, divided by the total time spent on either side, mean bias = 0.33). In contrast, prior to goal-cueing, two animals spent more time on the uncued side of the stem (mean bias = 0.10, see *Table 1* for results for individual animals). Moreover, during GOAL-CUE all animals also spent more time looking towards the cued arm than the uncued arm (mean bias = 0.13), again this bias was not observed prior to goal-cueing when only one animal spent more time looking towards the cued arm (mean bias = −0.11, see *Table 1* for results of individual animals). Furthermore, during RUN2 when the barrier was first removed and the food cue was no longer present, all four animals initially turned towards the cued arm and spent more time on the cued rather than the uncued arm (mean bias = 0.96, see *Table 1* for results for individual animals).

Consistent with the behavioural bias, in REST2 we found significant preplay of the yet unvisited cued arm (7.37% preplay events, p < 0.001, binomial test vs chance, *Figure 2A,D*). Conversely, the uncued arm was not significantly preplayed (4.41% preplay events p = 0.33, vs cued arm: p < 0.001, *Figure 2B,D*). Similarly significant effects were found when animals were analysed individually (see *Figure 2D*, *Table 2*), although the results for one animal were based on a relatively small sample (number of cued preplay events = 9, R1838), it still showed significant preplay of the cued arm. Moreover, the results were corroborated by a distribution-based analysis; namely, comparing the area under the curve (AUC) of bootstrapped cumulative distributions of absolute correlations for each arm to that of their shuffle distribution (*Figure 2A,B*, cued: p < 0.001, uncued: p = 0.22, cued vs uncued:

**Table 1.** Experimental parameters

|  | R1838 | R505 | R584 | R504 | All rats (mean) |
|---|---|---|---|---|---|
| **Cue bias (dwell time)** | | | | | |
| RUN1 | 0.33 | 0.32 | −0.05 | −0.20 | 0.10 |
| GOAL-CUE | 0.46 | 0.20 | 0.31 | 0.32 | 0.33 |
| **Cue bias (looking time)** | | | | | |
| RUN1 | −0.09 | −0.02 | 0.01 | −0.36 | −0.11 |
| GOAL-CUE | 0.06 | 0.31 | 0.09 | 0.05 | 0.13 |
| **RUN2 arm bias** | | | | | |
| RUN2 | 1.0 | 0.84 | 1.0 | 1.0 | 0.96 |
| **Session duration (min)** | | | | | |
| SLEEP1 | 60 | 88 | 75 | 74 | 74 |
| RUN1 | 13 | 10 | 13 | 9 | 11 |
| GOAL-CUE | 10 | 17 | 12 | 11 | 13 |
| SLEEP2 | 60 | 67 | 71 | 60 | 65 |
| RUN2 | 34 | 19 | 35 | 31 | 30 |
| **Template length (number of cells)** | | | | | |
| Up cued arm | 20 | 36 | 53 | 43 | 38 |
| Down cued arm | 15 | 35 | 45 | 33 | 32 |
| Up uncued arm | 19 | 26 | 45 | 40 | 33 |
| Down uncued arm | 15 | 32 | 41 | 43 | 33 |

p = 0.0044, *Figure 2—figure supplement 1*). Preplay events of the cued arm were equally likely to represent paths *to* and *from* the cued arm (7.34% vs 7.40% p = 0.49) and to run towards ('forward') and away ('reverse') from the ends of the arms (6.93% vs 7.80%, p = 0.18, *Figure 3A*). Moreover, the amount of preplay exhibited by each animal appeared to be predicted by the interest they displayed for the cued arm during GOAL-CUE (*Figure 3—figure supplement 3*). Importantly, preferential preplay of the cued arm could not be explained by differences in the number of cells with fields on the arms (*Figure 3—figure supplement 2A–B*), spike-sorting quality (cells with neighbouring place fields were as well separated in cluster-space as those with distant fields, p = 0.45, 2-sample Kolmogorov–Smirnov test), place field stability on the two arms (cued arm stability r = 0.54 vs uncued arm stability r = 0.49, p = 0.15) or the location of place fields on the cued arm (*Figure 3B*, p = 0.22 two-sample Kolmogorov–Smirnov test). In sum, we found during rest after goal-cueing, significant and preferential preplay of an unvisited and motivationally relevant portion of the environment.

Does goal-cueing trigger preplay? If so, there should be a greater number of significant pre-play events in REST2 compared to REST1 which was recorded before animals had visited or seen any part of the environment. Preplay of the cued arm was higher in REST2 than REST1 (7.37% vs 4.74%, p < 0.001, *Figure 2C,E*), an effect that was seen for all animals (*Table 2*). Indeed, the cued arm was not significantly preplayed during REST1 (4.74%, p = 0.34). Again, the result was corroborated using an AUC analysis (*Figure 2C*, *Figure 2—figure supplement 1*). Thus, we find preplay only occurs during rest periods recorded after goal-cueing. However, it is possible that the frequency of preplay might decrease as a function of the temporal gap between rest and behaviour. As such our failure to detect preplay in REST1 might be due to the greater delay between REST1 and RUN2 than between REST2 and RUN2. To address this we analysed preplay of the stem (i.e., RUN1) during REST1. We did not find preplay of the stem (4.12% preplay events, p = 0.44, AUC analysis p = 0.053, *Figure 2—figure supplement 2*, *Table 3*, RUN1 REST1 vs cued REST2: p < 0.001). Consequently, these results imply that the preplay of the unvisited, yet visible, environment we observed in REST2 was driven by behavioural cueing of that environment.

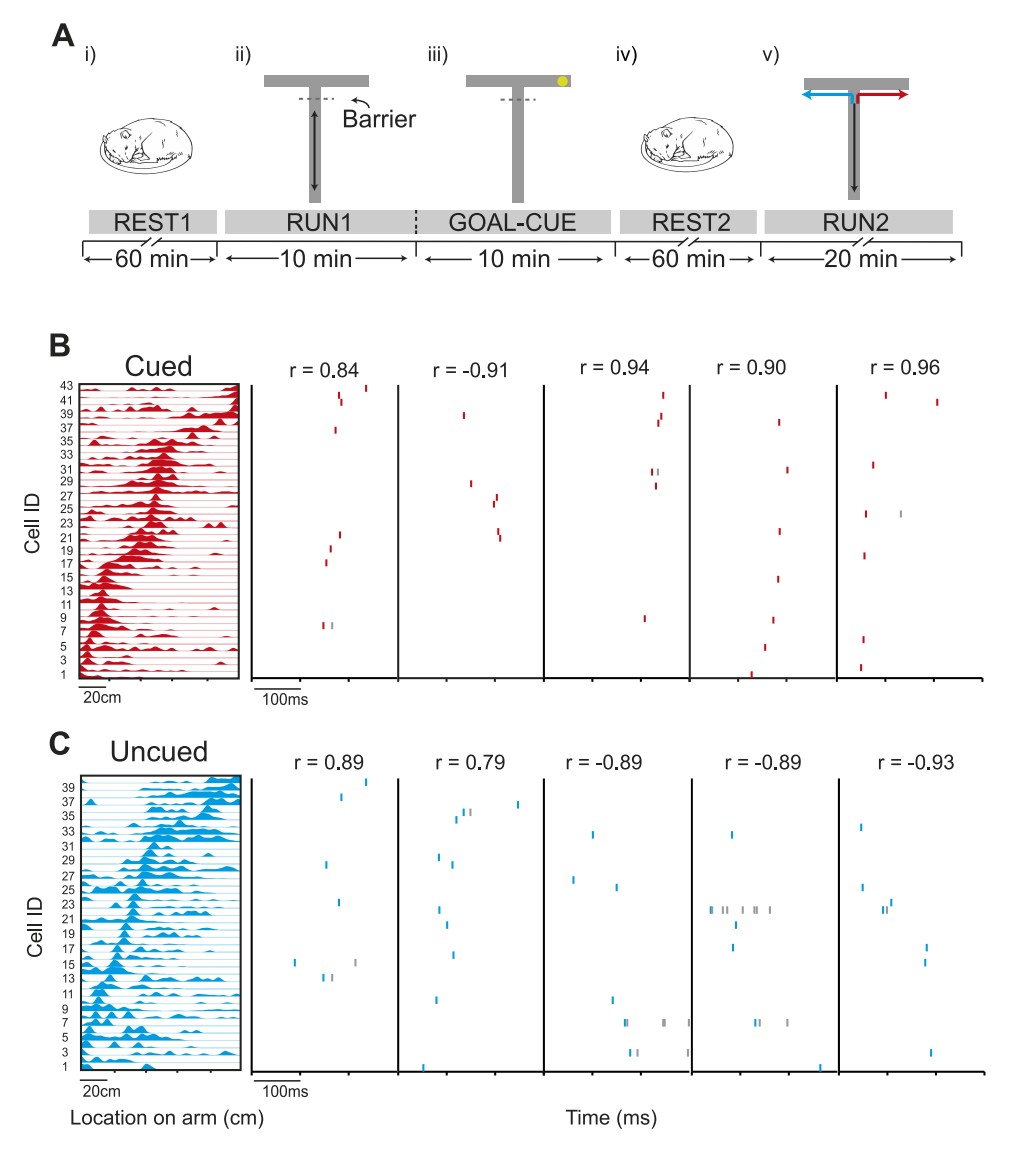

**Figure 1**. Preferential preplay of a behaviourally relevant, unvisited environment. (**A**) Experimental protocol. (**i**) Prior to running on the track, the animals rested for at least an hour (REST1). (**ii**) Following REST1, animals ran 20 laps on the stem (RUN1). Access to the arms was blocked by a barrier at the end of the stem which the animals could see through but not pass. (**iii**) Following RUN1, the experimenter baited one arm so to provoke the animals' interest in that arm (GOAL-CUE). (**iv**) Following goal-cueing, the animal rested for at least another hour (REST2). (**v**) Following REST2, the barrier was removed and the animals traversed the extent of the track, in alternate L-shaped laps (RUN2). (**B**) Left: an example template for a run *to* the cued arm. x-axis shows location on the track and y-axis cell IDs. Right: Example raster plots of preplay events—the title shows the correlation between the preplay event and the template sequence. **C** same as **B** but for the uncued template.

The following figure supplements are available for figure 1:

**Figure supplement 1**. Schematic of experimental apparatus black rectangles on track represent texture cues, dotted line transparent barrier, circle rest enclosure (18 cm wide), wiggly border demarcates the white curtains surrounding the environment, with distal landmarks fixed to the curtain ('X' and '0').

**Figure supplement 2**. Ripple power is elevated during preplay events power in the ripple spectrum (80–250 Hz) is higher during preplay events (red) than during both non-significant spiking events (black, p < 0.001) and non-event periods (dashed line, p < 0.001).

*Figure 1. continued on next page*

*Figure 1. Continued*

**Figure supplement 3**. Place cell templates place cell sequences for each template.

**Figure supplement 4**. Spiking events for cued arm in REST2 centre: bootstrapped cumulative distribution of (absolute) correlations between spiking events and the cued template in REST2 (red = data, black = shuffle).

At what point does preferential preplay of the cued arm emerge? Plausibly preplay might be initiated immediately when the cued arm is baited (start of GOAL-CUE) and simply persist into the subsequent REST2 period, alternatively the bias may only arise during rest. Due to the short duration of the goal-cueing period (~10 min) a relatively small number of spiking events were recorded for the two arms during this period (172 and 170 for the cued and uncued arm respectively). However, based on a bootstrapped comparison of the AUC for absolute correlations from the cued and uncued arm vs shuffled distributions, we found that the cued but not the uncued arm was preplayed (p = 0.02, p = 0.24 respectively, *Figure 3C*, *Figure 3—figure supplement 1*). A direct comparison of the proportion of preplay events for the cued vs uncued arm was marginally not significant (6.4% vs 4.12%, p = 0.052, see *Table 4* for results for individual animals). Finally, to validate the results from this smaller dataset we carried out a further, more inclusive, analysis. Specifically, we tracked the temporal evolution of the bias in preplay by comparing the activity of cells from the cued and uncued arms at different points during the experiment. For every spiking event we computed the mean rate for cells that would subsequently have fields on the cued arm compared to those with fields on the uncued arm. During REST1 and RUN1 the future cued and uncued arm cells did not differ in activity, this was true for both the first and second half of these periods (mean cued/uncued rate ratio: REST1 early ratio = 0.96, p = 0.88, REST1 late ratio = 1.04 p = 0.09, RUN1 early ratio = 1.09 p = 0.32, RUN1 late ratio = 1.18 p = 0.22, *Figure 3D*). However, during GOAL-CUE cued arm cells were significantly more active than uncued arm cells, an effect that was most pronounced during the first half (5 min) of the cueing period (GOAL-CUE early ratio = 1.78, p = 0.01, late = 1.46, p < 0.01). Finally, the difference between the two groups persisted through the subsequent rest period, albeit attenuating with time (REST2 early ratio = 1.30 p < 0.001, REST2 late ratio = 1.10 p < 0.01, *Figure 3D*). Importantly, control analyses showed that the bias in sequential preplay is not a mere product of differing activity levels of the cells for the two arms (*Figure 3—figure supplement 2C–D*). Together, these findings indicate that biased pre-activation of future experiences is instantiated at the point when an environment becomes motivationally-relevant.

Finally, to corroborate the results obtained from the rank-order analysis of spike sequences, we applied a Bayesian spatial reconstruction algorithm (*Davidson et al., 2009*; *Bendor and Wilson, 2012*) to the data from the two rest sessions. In contrast to the rank-order method, which utilised only the first spike emitted by each cell, the Bayesian decoding approach used all spikes emitted during an event. The Bayesian decoding approach uses the spiking activity of all simultaneously recorded place cells to calculate the posterior probability of an animal being at any position in the environment, based on a Poisson spiking framework. (See Materials and methods; R1838 was excluded from this analysis due to low cell yield). The Bayesian decoder performed equally well on the cued and uncued arms (*Figure 4A,B*, median error for both arms = 10.0 cm). Next, we applied the Bayesian decoding to spiking events during the rest sessions. First, we calculated the posterior probabilities for 5 ms non-overlapping bins, which generated a posterior probability matrix for each event (*Figure 4C,D*). A spiking event that reflects a constant speed run through the environment will show an increased posterior along a line in the decoded posterior matrix. Therefore, for each spiking event we fit a line that accounted for the maximum variance and calculated its goodness of fit (*Figure 4C,D*). To assess if that event represented a significant preplay event: we generated 1000 posterior probability matrices by shuffling the identities of cells included in the event, fit lines on all matrices, and calculated their goodness of fits. Events whose goodness of fits exceeded the 95th percentile of the shuffled distributions were labelled as preplay events. Again, during REST2, we found preplay of the cued arm (7.64% of events, p < 0.001, binomial test) but not of the uncued arm (4.78% of events, p = 0.55, vs cued = p < 0.001, *Figure 4E*). Moreover, neither the cued nor the uncued arm were significantly preplayed in REST1 (cued = 5.04%, p = 0.43, uncued = 4.69%, p = 0.55, *Figure 4F*). Thus, this

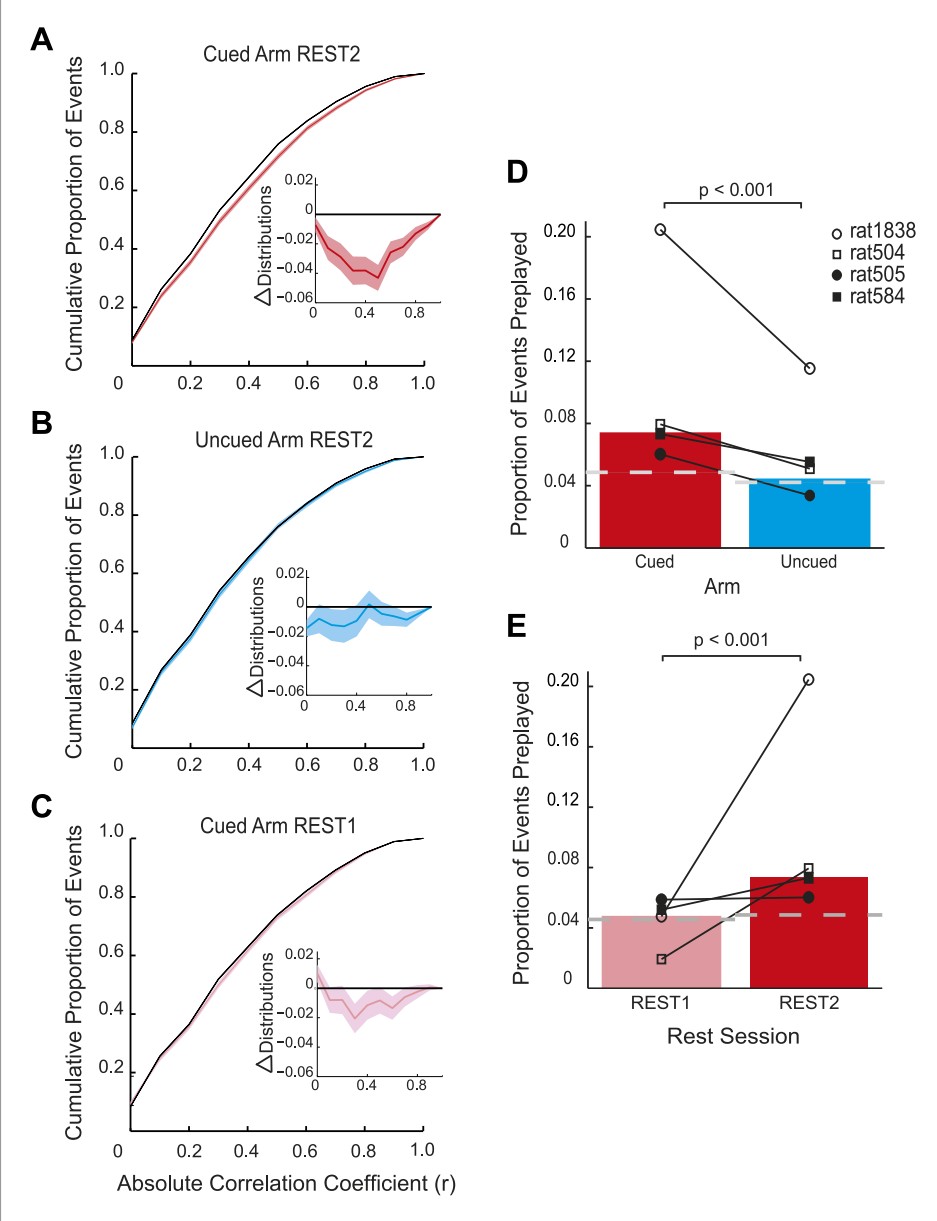

**Figure 2**. Preplay is a function of goal-cueing. (**A**) Bootstrapped cumulative distribution of (absolute) correlations between spiking events and the cued template in REST2 (red = data, black = shuffle). Lighter areas of the curve show 1 standard deviation of the mean. Inset: difference between the data and shuffle distributions. If there are more high correlations in the data compared to the shuffle then the data distribution will deviate below the shuffle distribution. (**B–C**) same as **A** but for the uncued template in REST2 and the cued template in REST1, respectively. (**D**) Proportion of spiking events categorised as preplay events in REST2 for the cued and uncued arms. Bars show mean for all animals, and the black lines show the result for each animal. The grey dashed line shows the proportion of preplay events expected by chance. (**E**) Same as **D** but comparing proportion of preplay events for the cued template in REST1 and REST2.

The following figure supplements are available for figure 2:

**Figure supplement 1**. Preplay of cued arm in REST2—distribution-based analysis **A** mean difference between the bootstrapped cumulative distributions of absolute correlations for the data and the bootstrapped shuffle for the cued and uncued arms in REST2 and the cued arm in REST1.

**Figure supplement 2**. Preplay of stem during REST1 bootstrapped cumulative distribution of absolute correlations for spiking events of the stem, recorded during REST1.

**Table 2**. REST period results

| Animal | Arm | # spiking events | # preplay events | % preplay | % chance | p-value |
|---|---|---|---|---|---|---|
| **REST2** | | | | | | |
| **R1838** | Cued | 44 | 9 | 20.45 | 6.86 | **$6.56 \times 10^{-4}$** |
| | Uncued | 26 | 3 | 11.54 | 6.77 | 0.10 |
| **R505** | Cued | 631 | 38 | 6.02 | 4.57 | **0.037** |
| | Uncued | 860 | 29 | 3.37 | 3.82 | 0.72 |
| **R584** | Cued | 437 | 32 | 7.32 | 4.73 | **$6.20 \times 10^{-3}$** |
| | Uncued | 398 | 22 | 5.53 | 4.74 | 0.19 |
| **R504** | Cued | 516 | 41 | 7.95 | 5.15 | **0.0027** |
| | Uncued | 373 | 19 | 5.09 | 4.40 | 0.21 |
| **All rats** | Cued | 1628 | 120 | 7.37 | 4.86 | **$4.12 \times 10^{-6}$** |
| | Uncued | 1657 | 73 | 4.41 | 4.22 | 0.33 |
| **REST1** | | | | | | |
| **R1838** | Cued | 63 | 3 | 4.76 | 7.05 | 0.66 |
| | Uncued | 35 | 2 | 5.71 | 7.37 | 0.48 |
| **R505** | Cued | 664 | 39 | 5.87 | 4.34 | **0.025** |
| | Uncued | 1215 | 38 | 3.13 | 3.45 | 0.70 |
| **R584** | Cued | 269 | 14 | 5.20 | 4.67 | 0.28 |
| | Uncued | 247 | 21 | 8.50 | 4.72 | **0.0035** |
| **R504** | Cued | 311 | 6 | 1.93 | 4.44 | 0.99 |
| | Uncued | 173 | 11 | 6.36 | 4.21 | 0.063 |
| **All rats** | Cued | 1307 | 62 | 4.74 | 4.56 | 0.34 |
| | Uncued | 1670 | 72 | 4.31 | 3.80 | 0.12 |

Summary results from REST1 and REST2 for the cued and uncued arms for individual animals. # Spiking events = total number of spiking events recorded. # preplay events = number of significant spiking events. % preplay = Proportion of the spiking events that qualified as preplay events (i.e., that were significant), expressed as a percentage. % chance = proportion of spiking events from the shuffled data that qualified as preplay events, expressed as a percentage. p-value = probability, derived from a binomial test, of obtaining the observed number of preplay events for each template given the chance level calculated from the shuffled data.

alternative, more powerful, analysis replicates the exact same pattern of results as demonstrated by our main analysis; indicating that goal-triggered pre-activation of future place cell sequences is a robust and reliable phenomenon.

## Discussion

Preplay of a visible, unvisited environment that has motivational relevance to an animal accords with various reports from neuropsychological and imaging studies indicating a crucial role for the hippocampus in processing future goals (*Spiers and Barry, 2015*), imagination, and future thinking (*Hassabis et al., 2007*; *Schacter et al., 2012*). Moreover, these findings agree with previous studies showing *replay* can be modulated by reward (*Cheng and Frank, 2008*; *Singer and Frank, 2009*; *Pfeiffer and Foster, 2013*) but extend this modulatory influence to preplay of future, goal-oriented routes. In our data, sequences of place cells leading *to* and *away* from the goal were preplayed with similar frequency and occurred in both forward and reverse directions (i.e., positive and negative correlations). Replay of sequences leading towards goal locations has previously been linked with the planning of future trajectories (*Pfeiffer and Foster, 2013*). Plausibly the elevated proportion of preplay events we observed leading to the cued goal might also reflect route planning—especially given the animals' subsequent preference for the cued arm. However, preplay of sequences leading from the cued goal towards the stem are less easily explained. On one hand it is possible that these

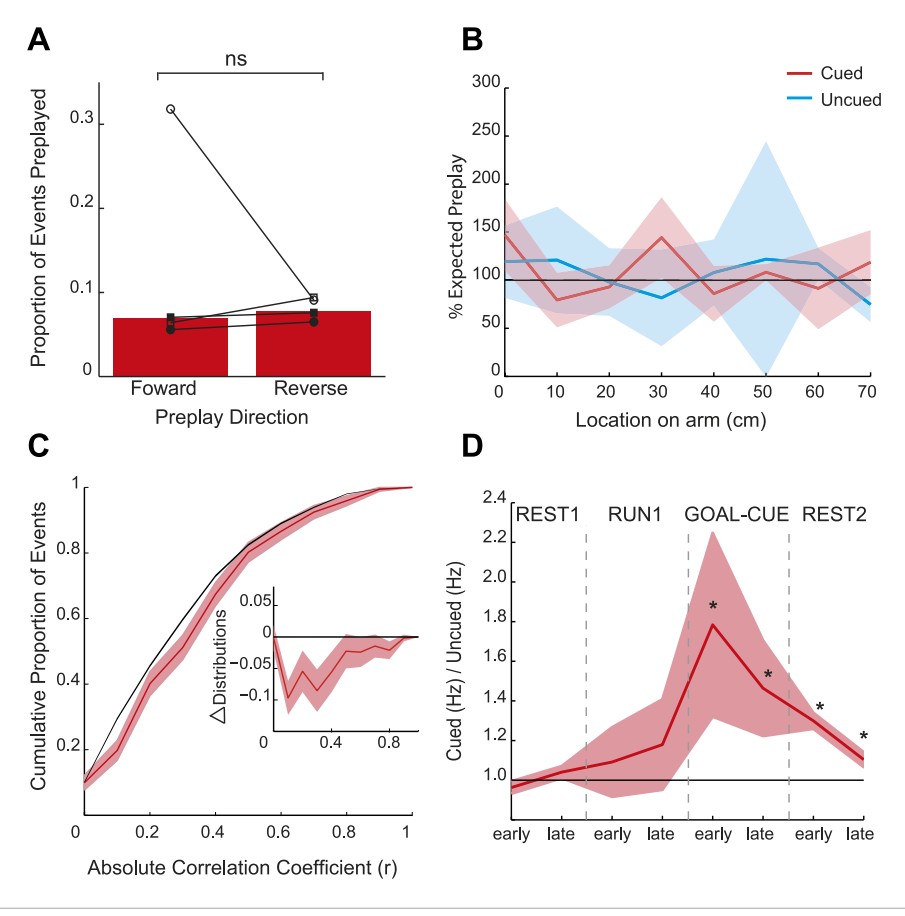

**Figure 3**. Spatial and temporal dynamics of preplay. (**A**) The proportion of preplay events when negative ('reverse') and positive ('forward') spiking event correlations are analysed separately. Bars show means for all data and black lines the results for each animal. (**B**) Frequency of preplay events vs location on the cued (red) and uncued (blue) arms normalised by the density of place field centres—100% indicates the expected number of preplay events under an even distribution across each arm. No bias towards particular sections of the arms was evident (cued p = 0.22, uncued p = 0.15, based on a two-sample Kolmogorov–Smirnov test). Lighter areas show standard error of the mean (SEM) and the black line the expected distribution. (**C**) Bootstrapped cumulative distribution of (absolute) correlations between spiking events and the cued template during GOAL-CUE (red = data, black = shuffle). Lighter areas of the curve show 1 standard deviation of the mean. Inset: difference between the data and shuffle distributions. (**D**) Ratio of activity levels between cued and uncued arm cells (cued/uncued) during events for the first and second half of each experimental period. Red line shows mean ratio, derived from bootstrapped data, obtained for each period, and the shaded areas 1sd of the bootstrapped data. The black horizontal line indicates equal rates for the two arms. * = significantly different from 1 based on 95% confidence intervals.

The following figure supplements are available for figure 3:

**Figure supplement 1**. Preplay of cued arm during GOAL-CUE—distribution-based analysis **A** mean difference between the cumulative distributions of absolute correlations for the bootstrapped data and shuffle for the cued and uncued arms during GOAL-CUE.

**Figure supplement 2**. Preferential preplay of the cued arm is not confounded by the number and activity of cells on the cued arm.

**Figure supplement 3**. Animals' interest in cued arm during GOAL-CUE is associated with subsequent preplay for each animal.

**Figure supplement 4**. Tetrode tracts example histology showing the location of tetrode recording from area CA1 in rat R504.

**Table 3**. REST1 stem results

| Animal | # spiking events | # preplay events | % preplay | % chance | p-value |
|---|---|---|---|---|---|
| R1838 | 68 | 3 | 4.41 | 6.01 | 0.59 |
| R505 | 980 | 35 | 3.57 | 4.03 | 0.74 |
| R584 | 329 | 17 | 5.17 | 4.53 | 0.24 |
| R504 | 396 | 18 | 4.55 | 3.50 | 0.11 |
| All rats | 1773 | 73 | 4.12 | 4.08 | 0.44 |

Summary results from REST1 analysing preplay of the stem. # Spiking events = total number of spiking events recorded. # Preplay events = number of significant spiking events. % preplay = Proportion of the spiking events that qualified as preplay events (i.e., that were significant), expressed as a percentage. % chance = proportion of spiking events from the shuffled data that qualified as preplay events, expressed as a percentage. p-value = probability, derived from a binomial test, of obtaining the observed number of preplay events for each template given the chance level calculated from the shuffled data.

sequences simply reflect planning for a return trip to the familiarity of the stem after retrieval of the food. However, reverse replay has often been characterised as a solution to the temporal credit assignment problem (*Foster and Wilson, 2006*; *Foster and Knierim, 2012*)—providing a learning mechanism to evaluate a sequence of actions that led to a successful outcome. In the context of preplay, when the cued arm has not been physically visited, an intriguing possibility is that paths towards the goal might be simulated using forwards preplay then evaluated and learnt from during reverse preplay, in a manner similar to that proposed to exist for periods of exploration and post-exploration rest (*Foster and Wilson, 2006*; *Diekelmann and Born, 2010*; *Buhry et al., 2011*).

Recently several studies have described 'de novo preplay'—preplay of environments that have yet to be experienced (*Dragoi and Tonegawa, 2011*, *2013*, *2013*)—which is believed to reflect the activity of preconfigured cell ensembles (*McNaughton et al., 1996*) that subsequently become bound to sensory cues in the environment. We did not observe de novo preplay in the current experiment (no preplay of stem during REST1). Although the origin of this discrepancy is unclear, we note that several other studies also do not find de novo preplay for a completely novel environment (*Wilson and*

**Table 4**. GOAL-CUE results

| Animal | Arm | # spiking events | # preplay events | % preplay | % chance | p-value |
|---|---|---|---|---|---|---|
| R1838 | Cued | 5 | 0 | 0 | 7 | 0.30 |
| | Uncued | 5 | 0 | 0 | 7 | 0.30 |
| R505 | Cued | 45 | 3 | 6.67 | 4.02 | 0.11 |
| | Uncued | 48 | 1 | 2.08 | 3.57 | 0.52 |
| R584 | Cued | 111 | 8 | 7.21 | 4.81 | 0.087 |
| | Uncued | 112 | 5 | 4.46 | 4.82 | 0.46 |
| R504 | Cued | 11 | 0 | 0 | 4.36 | 0.39 |
| | Uncued | 7 | 1 | 14.29 | 5.00 | 0.044 |
| All rats | Cued | 172 | 11 | 6.40 | 4.64 | 0.11 |
| | Uncued | 170 | 7 | 4.12 | 4.49 | 0.50 |

Summary results from GOAL-CUE analysis # Spiking events = total number of spiking events recorded. # Preplay events = number of significant spiking events. % preplay = Proportion of the spiking events that qualified as preplay events (i.e., that were significant), expressed as a percentage. % chance = proportion of spiking events from the shuffled data that qualified as preplay events, expressed as a percentage. p-value = probability, derived from a binomial test, of obtaining the observed number of preplay events for each template given the chance level calculated from the shuffled data.

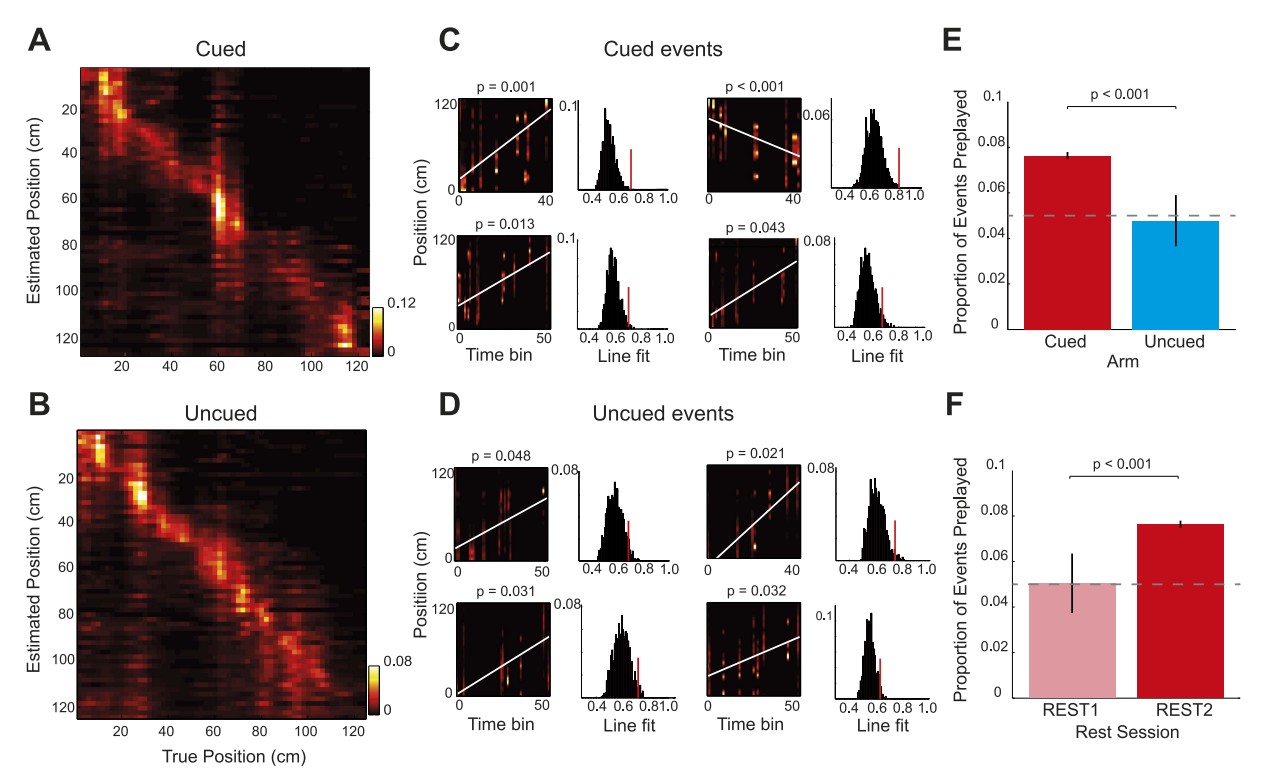

**Figure 4.** Bayesian reconstruction confirms preferential cued arm preplay. (**A–B**) Confusion matrices based on RUN2 data showing decoding accuracy for the cued (**A**) and uncued arms (**B**). Columns show the mean posterior probability distribution across the arm ordered by the true position of the rat (2 cm bins, data for R504). Decoding errors appear as power away from the identity diagonal—here both arms yield accurate estimates of the animal's position at all points on the track. (**C–D**) Representative preplay events for the cued (**C**) and uncued (**D**) arms showing, (left) the straight line trajectory that best fits the decoded event (x-axis indicates time bin (5 ms) within the event, y-axis position on arm, title indicates probability of obtaining a better trajectory by chance) and, (right) the null distribution of trajectory fits obtained by shuffling the cell identities for each event (x-axis, quality of trajectory fit, y-axis, proportion of shuffled events). (**E**) Proportion of spiking events categorised as preplay events for the cued and uncued arms in REST2. Bars show mean for all data with SEM. (**F**) Same as **E** but comparing preplay in REST1 and REST2 for the cued arm.

*McNaughton, 1994*; *Lee and Wilson, 2002*). However, it seems likely that other environmental factors present here but absent in the do novo preplay studies (*Dragoi and Tonegawa, 2011*, *2013*, *2014*), such as the barrier, and visible but inaccessible arms, might have ameliorated this phenomenon. Indeed, the AUC analysis comparing spiking events recorded during REST1 against the stem template (RUN1, *Figure 2—figure supplement 2*) was narrowly not significant (p = 0.053), suggesting that perhaps with more data or a simpler environment we might have found de novo preplay for the stem. In a similar vein one might also have expected to observe a background rate of de novo preplay for the uncued arm during REST2. Again it seems possible that the complexity of the environment worked against such an effect in that the motivationally relevant cued arm was preplayed at the expense of the uncued arm. What remains to be understood is if this bias reflects an active process, simulating trajectories associated with the goal, or if the relevance of the cued arm simply allows it to more effectively capture preconfigured cell assemblies.

In sum, our data indicates that preplay of an unvisited environment can be initiated in response to viewing that environment if it is relevant to future goals—features that were absent in previous studies investigating the formation of hippocampal representations for novel spaces (*Wilson and McNaughton, 1994*; *Lee and Wilson, 2002*; *Rowland et al., 2011*). Furthermore, although our data does not provide direct support for de novo preplay it also does not argue against it, and is consistent with the hypothesis that the hippocampus supports the construction of novel spatial experiences as suggested by other authors (for example see *Dragoi and Tonegawa (2014)*).

These results are consistent with the notion that place cells form preconfigured assemblies or 'charts' (*McNaughton et al., 1996*)—an active but unassigned chart being associated with the cued arm during GOAL-CUE, but do not preclude the possibility that place cell firing is driven in a feed-forward manner (e.g., *Hartley et al., 2000*). We conclude that these findings support the hypothesised role of preplay for preparation for future experiences (*Diba and Buzsaki, 2007*; *Erdem and Hasselmo, 2012*; *Foster and Knierim, 2012*; *Pfeiffer and Foster, 2013*) but extend it to simulating future experiences in environments yet to be actively explored. It remains to be seen if the ability of the hippocampus to construct future spatial sequences is related to the generation of more abstract temporal associations (*Kraus et al., 2013*; *Eichenbaum and Cohen, 2014*).

## Materials and method

### Animals and surgery

Four male lister-hooded rats were used in this study. All procedures were approved by the UK Home Office, subject to the restrictions and provisions contained in the Animals (Scientific Procedures) Act of 1986. Two rats (330–333 g at implantation) received two microdrives, each carrying four tetrodes of twisted 25 μm HM-L coated platinum iridium wire (90% and 10%, respectively; California Fine Wire), targeted to the left and right CA1. One animal (350 g) received a single microdrive, carrying eight tetrodes, targeting right CA1. Finally, one animal (425 g) received one microdrive carrying four tetrodes of 12- μm wire targeted to the mEC and one carrying four tetrodes of 17- μm wire targeted to contralateral CA1. Only single units from the CA1 implant were included in this study. The 12 and 17- μm wires were platinum-plated to reduced impedance to 200–300 kΩ at 1 kHz. Rats were maintained at 90% of free-feeding weight, and were housed individually on a 12-hr light/dark cycle. After surgery, rats were food deprived with *ad libitum* access to water.

### Recording

Screening was performed post-surgically after a 1-week recovery period. An Axona recording system (Axona Ltd., UK) was used to acquire the single-units, and positional data (for details of the recording system see Barry et al. [*Barry et al., 2007*]). The position and head direction of the rats were captured using an overhead video camera to record the position using a light-emitting diode (LED) on the animals' head-stages (50 Hz). Tetrodes were gradually advanced over days until place cells were found.

### Experimental apparatus and protocol

We used a T-shaped track, raised 43 cm off the ground. The track consisted of 10 cm wide runways covered in black rubber, the stem was 222 cm long and the arms 200 cm from tip to tip. Animals were not habituated to running on linear tracks prior to the experiment. However, all animals had been trained to forage for rice in a square enclosure located in the same room but outside the curtained area. Initially, to prevent animals from accessing the arms, a barrier was placed on the stem 22 cm from the junction with the arms. The barrier consisted of a grill of vertical wooden bars spaced so the animals could easily see through but could not scale it. The apparatus was located in a curtained environment, and two distal landmarks were mounted on the curtain on either side of the maze for orientation. To encourage even distribution of place fields along the track, textured local cues were placed along the stem and arms. These five different cues were sheets of: black rubber, grey cardboard, black polypropylene, plaster tape, and sandpaper each approximately 20 cm long and 10 cm wide. For rest periods, the animal was placed in a cylindrically shaped enclosure (18 cm diameter × 61 cm high) with a towel placed in the bottom. Prior to the experiment the animal was familiarised with the rest enclosure for a minimum of 120 min. The rest enclosure was located beside the track during rest periods, but was not present during the other periods.

Prior to running on the track, animals were placed in the rest enclosure for at least 60 min—REST1 (if the animals rested for less than 40 min during a session then the session was extended in order to obtain at least 40 min worth of recording). During this period the animals' quiescence was assessed based on speed estimates, derived by measuring the movement of the LED attached to the animals' headstages. Following REST1, RUN1 was initiated by placing the rats at the end of the stem facing away from the track. During RUN1 animals were encouraged to run up and down the stem by the experimenter rewarding alternate ends of the stem with rice grains. During this period animals were

prevented from running onto the arms by the barrier. Once animals had completed 20 laps, the experimenter stopped baiting the stem and instead baited the end of one of the arms (GOAL-CUE), which was inaccessible yet visible to the animal (cue located ca. 1 m from the animal). The cued arm was counterbalanced between animals. The animals had good visual access to the inaccessible arms as they could see between the wooden dowels of the barrier. The experimenter stood by the end of the cued arm to elicit further interest. Animals were allowed to remain on the track for at least a further 10 min (GOAL-CUE).

Following RUN1, the animals were placed back into the rest enclosure for at least 60 min—REST2. Following REST2, animals were put back on the end of the stem facing away from the track. The barrier at the end of the stem was removed and the animals were allowed to traverse both the stem and arms in alternate L-shaped laps for each arm—RUN2. On the first lap during RUN2, animals were allowed to choose which arm they turned to when reaching the end of the stem. On subsequent laps animals were forced to alternate—a barrier (the same as the one used for RUN1) was placed at the junction of the T-maze to occlude each arm in turn, and was moved between trials by the experimenter. The animals completed 20 L-shaped laps for each arm.

## Data analysis

### Template generation

Ratemaps for the runs on the arms during RUN2 were generated. Prior to ratemap generation, the last 20 cm (10%) of each arm were removed to exclude areas where the animal groomed, ate etc. Similarly, to avoid non-perambulatory behaviours, position samples where the animal's speed was lower than 3 cm/s were excluded. Animals' paths were linearised and dwell time and spikes binned into 1 cm bins, then smoothed with a Gaussian kernel ($\sigma$ = 2 cm). Firing rate was calculated for each bin by dividing spike number by dwell time. Templates were generated based on the location of the peak firing rate of each cell (if a cell had more than one place field, the field with the highest firing rate was used). Specifically, four templates were generated, two for each arm—one for runs commencing at the start of the arm (i.e., adjacent to the stem) and concluding at the far end of the arm (TO cued/uncued arm) and vice versa (FROM cued/uncued arm). These templates we refer to as Up Cued Arm (UCA), Down Cued Arm (DCA), Up Uncued Arm (UUA) and Down Uncued Arm (DUA). For example, the Up Cued Arm (UCA) template enumerates the order in which the peaks of firing fields appear on the cued arm for runs towards the end of the arm. So, if cells with IDs 3, 20 and 7 were the first three to have peaks on that arm the UCA template would start 3, 20, 7. Cells whose peak firing rate in the linearised ratemap were below 0.5 Hz, had less than five contiguous bins with rates above the mean firing rate of the cell, or whose spatial correlation between the first and second half of the RUN2 session was less than 0.3 were excluded.

For a follow-up analysis we generated templates for the stem based on the RUN1 session. Templates were generated in the same way as for the arms.

## Preplay analysis

For each rest period, times where at least 15% of cells from a given template fired within 300 ms and were bound by at least 50 ms of silence were selected as 'spiking events' (for R1838, which had a lower cell yield than the other rats, a minimum of 4 cells were required to be active). If a single cell fired more than one spike within this period, the first spike was counted and other spikes disregarded. The extent to which templates were represented in the spiking events was assessed using template-matching (*Lee and Wilson, 2002*; *Foster and Wilson, 2006*; *Diba and Buzsaki, 2007*). Specifically, the temporal sequence of cells in spiking events and the order of their corresponding peaks in the template was compared. If future template sequences are preplayed in rest the order in which cells fire during spiking events would be expected to resemble the order in which they fire on the track. In other words, one would expect the spike sequence in events and in future template sequence to be more correlated than predicted by chance. To determine if each template was being significantly preplayed we compared the proportion of spiking events that individually correlated with that template with the proportion expected by chance (as predicted by a shuffling procedure). Specifically, the Spearman's rank-order correlations between each spiking event and the current template was computed and the number yielding significant correlations (based on a two-tailed test, each tail tested at the 97.5% level) was recorded; these were labelled preplay events. Both positive as well as negative

significant correlations were categorised as preplay events, as previous studies have reported that replay can reactivate a place cell sequence in the forward or reverse order (*Lee and Wilson, 2002*; *Diba and Buzsaki, 2007*). Next, the sequence of spikes in each event was randomly permuted (shuffled) 100 times by randomly re-assigning cell IDs such that spikes from the same cell were never separated. For each permutation the Spearman's rank-order correlation and matching p-value were calculated, again the proportion of significant correlations was recorded; this value represented the proportion of preplay events expected by chance. Finally, the proportion of shuffled events that had a significant correlation with the future template sequence was compared to the number obtained for the unshuffled data using a one sample binomial test. To assess whether the cued arm was preplayed more than the uncued arm we directly compared the total number of events and number of preplay events, again using a binomial test. A similar approach was used for comparing number of preplay events in REST1 and REST2.

Subsequently, we corroborated these results using an alternative- distribution-based, analysis. Namely, we assessed whether the AUC of the cumulative distribution of absolute correlations for each arm in each of the two rest sessions differed from their shuffle distribution. Specifically, for each arm and each period, we bootstrapped the population of absolute correlations obtained between templates and spiking events 10,000 times. On each iterations we computed the difference between the AUC of the cumulative distribution of the bootstrapped data vs the AUC of the bootstrapped shuffled data correlations, which had been obtained previously. 95% confidence intervals (2.5% and 97.5% percentiles used, i.e., a two-tailed test) were determined from these difference scores and were used to assess whether the two distributions differed significantly. Specifically, if the interval did *not* contain 0 then we concluded the two distributions were significantly different.

To analyse preplay during GOAL-CUE we performed the same analysis for spiking events emitted while the animal was on the stem. Specifically, we only considered events recorded when an animal's velocity was below 10 cm/s and the animal was located within 20 cm (10% of track length) of the barrier.

As an additional control for differences in the number of cells contributing to each template we repeated the analyses described above after first down-sampling each template to equate the number of cells (i.e., to match the length of the shortest template). Specifically, if the shortest template consisted of 20 cells while another template consisted of 30 cells, we randomly removed 10 cells from the longer template. Moreover, we repeated this down-sampling process a 100 times, each time removing a random set of 10 cells. We then ran the preplay analysis as before—estimating the number of preplay events for each down-sampled cued and uncued template and comparing them to each other and chance levels. A similar procedure was used to equate activity rates for cued and uncued templates during the two rest periods. Namely, the distribution of events, containing different numbers of cells, were equated for the different templates. For example, if one template had 50 events with 5 cells active and another only had 40 events, we randomly removed 10 events from the former template. This process was then repeated for events of different lengths. As before the entire down sampling procedure was repeated 100 times. As well as controlling for quiescent rate differences this analysis also controls for the number of events for each template.

To analyse the temporal evolution of preplay across the entire experiment we used an alternative approach to the template-matching described above. This was done due to the short duration of the RUN1 and GOAL-CUE periods (~10 min), which limited the power of the template-matching approach. As an alternative, we computed the mean rate of future cued and uncued arm cells during spiking events for each half of all experimental periods (i.e., REST1, RUN1, GOAL-CUE and REST2). If a cell was active on both arms we excluded it, unless the rate on one arm was at least twice as high as the rate on the other arm. To assess statistical significance of rate differences we bootstrapped the event rates for the cued and uncued arm cells 10,000 times. We then computed the mean ratio between the cued and uncued arm rates (cued/uncued) for each bootstrap, and estimated the 95% confidence interval. If the interval *did not* contain 1 then we deemed the rate difference significant.

## Bayesian reconstruction analysis

To corroborate results obtained from the rank-order correlation between spikes and field position we also applied a Bayesian reconstruction algorithm to spike sequences recorded during events. Following *Zhang et al., 1998*, and *Davidson et al., 2009*, using non-overlapping temporal windows,

we computed the probability distribution across position based on ratemaps generated from RUN2 (*Davidson et al., 2009*; *Zhang et al., 1998*).

Specifically during a time window ($T$) the spikes generated by $N$ place cells was $\boldsymbol{K} = (k_1, \ldots, k_i, \ldots, k_N)$, where $k_i$ was the number of spikes fired by the $i$th cell. The probability of observing $\boldsymbol{K}$ in time $T$ given position ($x$) was taken as:

$$P(\boldsymbol{K}|x) = \prod Poisson(k_i, T\alpha_i(x)) = \prod_{i=1}^{N} \frac{(T \times \alpha_i(x))^{k_i}}{k_i!} \times e^{-T\alpha_i(x)}, \tag{1}$$

where $x$ indexes the 1 cm spatial bins defined on the arms of the apparatus and $\alpha_i(x)$ is the firing rate of the $i$th place cell at position $x$, derived from RUN2 ratemaps.

To compute the probability of the animals' position given the observed spikes we applied Bayes' rule, assuming a flat prior for position ($P(x)$), to give:

$$P(x|\boldsymbol{K}) = R\left(\prod_{i=1}^{N} \alpha_i(x)^{k_i}\right) exp\left(-T\sum_{i=1}^{N} \alpha_i(x)\right), \tag{2}$$

where $R$ is a normalising constant depending on $T$ and the number of spikes emitted. Note we do not use the historic postion of the animals' to contstrain $P(x|\boldsymbol{K})$ thus the probability estimate in each $T$ is independent of its neighbours.

To validate the effectiveness of the reconstruction algorithm it was used to decode the animals' actual position during RUN2. For each $T$ (200 ms) location was decoded by taking the peak of the $P(x|\boldsymbol{K})$ distribution. Decoded estimates of location were compared with true location to obtain a measure of the absolute decoding error.

To decode offline activity, posterior probability matrices were produced for events with ≥7 active cells using 5 ms non-overlapping time windows ($T$). To decode the probability matrices we used a method similar to *Davidson et al. (2009)*, fitting a straight line—equivalent to a constant speed trajectory—to each matrix. Lines were defined with a gradient ($V$) and intercept ($c$), equivalent to the velocity and starting location of the trajectory. The goodness of fit of a given line ($R(V, c)$) was defined as the proportion of the probability distribution that lay within 20 cm of it. Specifically where $P$ is the probability matrix obtained from (2):

$$R(V, c) = \frac{1}{n}\sum_{t=0}^{n-1} P(|x(t) - (V.t.T + c)| \le d), \tag{3}$$

where $t$ index the time bins of width $T$ and $d$ is set to 20 cm. $R(V, c)$ was maximised using an exhaustive search to test all combinations of $V$ between $-50\ ms^{-1}$ and $50\ ms^{-1}$ in $0.2\ ms^{-1}$ increments (excluding slow trajectories with speeds $> -2\ ms^{-1}$ and $< 2\ ms^{-1}$) and $c$ between $-15$ m and 16 m in 0.01 m increments.

To assess the presence of preplay in each event we compared the goodness of fit of the best straight line trajectory with the best fits made to 1000 posterior probability matrices generated by shuffling the identities of cells included in the event. Events whose best fit line exceeded the 95th percentile of its own shuffled distribution we labelled as preplay events.

## Control analyses

To eliminate the possibility that poor cluster isolation might contribute to the observed preplay (such as mistakenly dividing a complex spike into two clusters), we analysed the cluster separation between every pair of place cells recorded from the same tetrode. Thus, for each spike the first three principle components of the waveform recorded on each of the four channels of the tetrode were calculated. Thus, 12 features (4 channels × 3 principle components) defined each spike. Based on these features the squared Mahalanobis distance between that spike and the centre of another cluster was calculated, using the covariance matrix for all spikes belonging to the cluster. This process was then repeated for all spikes in a given cluster and the mean Mahalanobis' distance calculated.

Clusters of spikes originating from a single cell but which have mistakenly been assigned to two or more cells will be identified by low a Mahalanobis distance and proximate place fields. We divided cell pairs into two groups—those with short Mahalanobis distances (less than the median) and those with long distances (greater than or equal to the median). The distribution of place field separation for the two groups was then compared using a 2-sample Kolmogorov–Smirnov test.

To eliminate the possibility that preplay of the cued arm during REST2 merely reflects familiarity with the proximal portions of the cued arm during GOAL-CUE we assessed whether off-line spikes

were more likely to be associated with the first half of the arm (i.e., closest to the stem) vs the second half (excluding the first 22 cm that are common to both arms). We computed the proportion of spikes representing proximal locations and assessed whether the proportion differed significantly (using a binomial test) from what one would expect based on the place field distribution of the cued arm. We also repeated the same analysis for the uncued arm. Finally, to assess whether the distribution of preplayed locations (normalised by place field distribution) for the two arms differed significantly from a uniform distribution we used a two-sample Kolmogorov–Smirnov test.

Finally, to assess place field stability on the cued and uncued arm we split the RUN2 session in half (based on odd and even samples) and generated ratemaps for each half. We then calculated the correlation between the two ratemaps (bin-wise Pearson product–moment correlation coefficients were calculated after first removing unvisited bins and any bin pairs that had mutual 0 Hz firing rates) and compared correlations obtained for the cued and uncued arms using a two-sample t-test.

## Local field potential analysis

LFP from CA1 was recorded at 4.8 kHz throughout the experiment. To analyse sharp-wave ripples (*O'Keefe and Nadel, 1978*; *Buzsaki et al., 1992*) the LFP was band-pass filtered between 80 and 250 Hz. An analytic signal was constructed using the Hilbert transform, taking the form:

$$s_a(t_k) = s(t_k) + iH[s(t_k)], \tag{4}$$

where H specifies the Hilbert transform, $s(t_k)$ is the filtered LFP signal, $t_k = k\Delta$, where $k = 1,...,K$ indexes the time-step and $\Delta$ is the inverse of the sampling rate.

An instantaneous measure of power was found by taking the squared complex modulus of the signal at each time point. This measure was then down sampled to 50 Hz to match the position sampling rate, and finally was smoothed with a boxcar filter of width 0.1 s. Two-tailed independent samples t-tests were used to compare power in the ripple-band during significant (i.e., preplay events) and non-significant spiking events and to compare power during significant spiking events and periods outside of spiking events.

## Histology

Rats were anaesthetised (4% isoflurane and 4 l/min $O_2$), injected intra-peritoneal with an overdose of Euthatal (sodium pentobarbital) after which they were transcardially perfused with saline followed by a 4% paraformaldehyde solution (PFA). Brains were carefully removed and stored in PFA which was exchanged for a 4% PFA solution in PBS (phosphate buffered saline) with 20% sucrose 2–3 days prior to sectioning. Subsequently, 40–50 µm frozen coronal sections were cut using a cryostat, mounted on gelatine-coated glass slides and stained with cresyl violet. Images of the sections were acquired using an Olympus microscope, Xli digital camera (XL Imaging Ltd.). Sections in which clear tracks from tetrode bundles could be seen were used to determine the location of cells recorded.

## Behavioral analysis

Animals' interest in the cued arm during goal-cueing was assessed as follows. The portion of the stem within 20 cm of the barrier was notionally divided in two long-ways to yield a cued and uncued side. Time spent on the two sides was determined based on the animals' tracked positions. Finally, to express interest in the cued arm as a single behavioural measure, the difference between the time spent on the cued vs the uncued side was divided by the total time spent on that section of the track. Thus, a value above 0 indicates that an animal spent more time on cued side than the uncued side. As an additional measure of the animals' interest in the cued arm during the goal-cueing period we assessed the amount of time animals spent looking toward the cued arm (while the animals' velocity was at least 5 cm/s, head direction was inferred by tracking a single LED which requires the animal to be moving in order to get accurate head direction estimates). Again, preference for the cued arm was expressed as a single behavioural measure by subtracting the amount of time spent with the head oriented towards the cued arm from the time spent with the head oriented towards the uncued arm and dividing by the total time spent with the head oriented towards either arm. Finally, to assess the animals' preference for the cued arm during the first lap of RUN2, the difference between the time spent on the cued arm vs the uncued arm during the first 3 s following the animals' entry onto the arms was divided by the total time spent on either arm.

## Acknowledgements

We thank Della Nicolle and Hannah Munsoor for help piloting the experiment, and Neil Burgess, Daniel Bendor and Peter Dayan for discussion and comments on a draft manuscript. Funding was provided by a Wellcome Trust Advanced Training Fellowship to HJS, a UCL studentship to HFO, a Wellcome Trust and Royal Society Fellowship to CB and a Sir Henry Wellcome Fellowship to DH.

## Additional information

### Funding

| Funder | Grant reference | Author |
| --- | --- | --- |
| Wellcome Trust | Advanced Training Fellowship | Hugo J Spiers |
| Wellcome Trust | Sir Henry Dale Fellowship | Caswell Barry |
| Royal Society | Sir Henry Dale Fellowship | Caswell Barry |
| University College London | Excellence Fellowship | Caswell Barry |

The funder had no role in study design, data collection and interpretation, or the decision to submit the work for publication.

### Author contributions

HFÓ, CB, Conception and design, Acquisition of data, Analysis and interpretation of data, Drafting or revising the article; HJS, Conception and design, Analysis and interpretation of data, Drafting or revising the article; ABS, Analysis and interpretation of data, Drafting or revising the article; DH, Conception and design, Drafting or revising the article

### Ethics

Animal experimentation: All procedures were approved by the UK Home Office, subject to the restrictions and provisions contained in the Animals (Scientific Procedures) Act of 1986.

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
