## [Decision Letter]

[Editors’ note: this article was originally rejected after discussions between the reviewers, but the authors were invited to resubmit after an appeal against the decision.]

Thank you for choosing to send your work entitled “The Hippocampus Constructs Desired Paths through Unexplored Space” for consideration at *eLife*. Your full submission has been evaluated by Eve Marder (Senior editor), a Reviewing editor, and three peer reviewers, and the decision was reached after discussions between the reviewers. Based on our discussions and the individual reviews below, we regret to inform you that this manuscript will not be considered for publication in *eLife*. Because *eLife* has a policy to not request new data collection at the revision stage and the reviewers felt considerable work was necessary, we are rejecting your manuscript. That said, the Reviewing editor and the reviewers thought your manuscript was sufficiently potentially interesting that they would be willing to consider a new submission of a revised manuscript, if it is responsive to the issues raised in this review, in particular to potentially include some new data collection. This invitation in no way should be taken as a guarantee that a new submission would be sent to the same reviewers or would meet with success.

The reviewers are consensual in their overall enthusiasm about the study and its findings. At the same time there were major concerns about the robustness of the main results and some additional issues:

1) According to Table 2, in one animal (R505) the number of sequences representing the cued locations are not significantly higher than chance. Moreover, in another animal (R1838) only 9 such sequences were detected so it seems that the recording was not optimal and this argues for excluding this latter animal from the dataset. Therefore, only 2 out of the 4 animals firmly support the conclusions of the work. Moreover, in the REST1 sessions in two animals these numbers were significant (one cued and one uncued arm). Therefore, more data are needed to convincingly demonstrate the conclusions of this work. At least showing it in one more animal to have n=3 showing the results could be sufficient.

The reviewers had the following recommendations to improve the robustness of the findings. One reviewer suggested that performing a sequence analysis animal-by-animal in the goal-cue run session would be more conclusive than it is in the REST2 session. In this case the claim could be related to prepay in the run session and this enough would be significant, not requiring the collection of more data.

Also, sequence analysis using Bayesian path prediction is more powerful than the currently used correlation analysis. Such analysis may provide more conclusive results for R505 as well in the REST2 session in which case further data may not be needed. This analysis should be pursued instead of or in addition to the correlation approach.

Another reviewer suggested that, given the small effect sizes, the authors should do additional analyses to validate their key results. There are two main analyses used to determine differences in preplay: first is a comparison of the proportion of significant preplay events. This method is sensitive to how the significance of each individual event is determined. The authors use rank correlation to compare the preplay event sequence to place cell templates in order to determine significance. The significance of each individual event should be determined by comparing to a distribution of correlations obtained by shuffling the cell order in the templates. Also, the significance value needs to be set at the p=0.025 level to control for multiple comparisons due to the two directions (Up and Down) of the run. How does this affect Figure 2? Further, rat1838 has very few events, and presents as an outlier in this analysis. For the second distribution-based analysis, the authors do not directly compare the Cued and Uncued arm r-value distributions. Also, raw distributions should be shown (not just cumulative distributions), and can be compared using the Kolmogorov–Smirnov test.

2) Another major concern speaks to evidence that the effect is functionally related to constructed representations of desired paths. Wording implying 'desired paths' may be an overstatement. The data do not speak directly to such a broad claim. It is also the case that there was as many forward and reverse sequence events detected. One might expect a preferential bias to forward event sequences if the function was representation of the unexplored path toward the reward. A strong recommendation would be to downplay the statements about constructed desired paths and also discuss the implication of reverse events being found in similar frequency, including alternative accounts like proposed by Foster and Wilson (2006).

3) The results contradict previous results on preplay, and this is not addressed in the manuscript. The conclusion that visual observation and goal cueing is necessary for observing preplay of a future experience contradicts previous results from Dragoi and Tonegawa (2011, Nature; 2013, PNAS; 2013, eLife) that show that preplay of a novel experience occurs without these conditions. While the lack of preplay of the uncued arm in the current study can be attributed to biasing of preplay towards a more motivationally relevant arm, the lack of preplay of the STEM arm during REST1 (Figure 2—figure supplement 2; and Table 3) directly contradicts previous results. The previous interpretation was that pre-existing temporal firing sequences in the hippocampus (cellular assemblies based on functional connectivity) become bound to novel environments as place cell sequences. But the current results force a re-interpretation that preplay of a novel environment only occurs if it is visually observed. This needs to be explicitly addressed in the manuscript, and requires a careful re-examination of the analysis.

[Editors’ note: what now follows is the decision letter after the authors submitted for further consideration.]

Thank you for resubmitting your work entitled “The Hippocampus Constructs Desired Paths through Unexplored Space” for further consideration at *eLife*. Your revised article has been evaluated by Eve Marder (Senior editor), a member of the Board of Reviewing Editors, and two peer reviewers. The manuscript has been improved but there are some remaining issues that need to be addressed before acceptance, as outlined below. Briefly, both of the peer reviewers felt that the recommended Baysian analysis is a requisite.

The authors have filled some of the missing gaps in the analysis based on the reviewers’ comments, and the manuscript is substantially improved. However, the concerns about the small effect sizes and the robustness of the results still remain. This is also apparent in the raw distributions that the authors now show in Figure 2—figure supplement 1 and Figure 3—figure supplement 1. A more-powerful Bayesian based decoding analysis (originally the suggestion of Reviewer 1) is necessary to confirm these results, and this will have a substantial impact on how these results will be received.

The correlation based template matching method that the authors have used in the manuscript is less effective than Bayesian decoding in detecting sequential activation/ reactivation (e.g. Carr et al., Nat. Neurosci. 2011, Vol. 14, pp 147–153). There is a concern that the rank correlations-based method may lead to false positive results due to other factor such as the burst propensity differences between cells. The fact that significant preplay events were seen in REST1 in one cued and another uncued arm is still hard to interpret. This may indeed show preplay as in the Dragoi studies but then why don't we see it in all animals? And why don't we see significant preplay of the uncued arms in REST2 too? Of course there is the other interpretation that significant preplay in REST1 may represent false positive detection, which may question how consistently one may see the preferential preplay of the cued arm (in REST2) if the experiments are repeated many times.

Bayesian decoding uses all of the spikes that occur during a reply event, and the entire place field structure to estimate the reactivated sequence of positions. In contrast, the correlation method compare the sequence of first spike of each cell during replay, and the relative location of place field centers, and is therefore less effective when only a subset of neurons participate in the replay event. Although the authors mention that they use multiple methods to compare the preplay of the cued arm to the uncued arm, the core method still remains the same, the correlation based template matching. Based on the output of this correlation-based analysis, the authors then evaluate the results using either the comparison of the number of significant events, or the distribution of the r-values. Comparing the distributions is a more powerful evaluation, and the differences between the cued and uncued arm preplays are small. Moreover, the difference in data and shuffle distributions are also small for Cued Arm preplay in Rest 2 (Figure 2—figure supplement 1). As a comparison, for previous results in de novo preplay, (Dragoi and Tonegawa, 2011, Nature), which the authors mention might be a relatively more subtle phenomenon, the preplay effect was stronger based on the distribution based analysis.

The authors have sufficient data to run a Bayesian based decoding analysis in at least 3 of the animals. Therefore, my strong recommendation would be to evaluate their data using this more powerful approach. Confirmation of their results using this method will substantially alleviate any concerns regarding the robustness of this effect, and make a stronger impact.

---

## [Author Response]

[Editors’ note: the author responses to the first round of peer review follow.]

We have been able to address the main concerns and suggestions raised, and have conducted numerous additional analyses all of which support our original conclusions. In light of the reviewers’ comments and the new analyses we have extensively revised our manuscript, which we now present for resubmission. In brief, the main changes are:

a) We clarify that we do see significant pre-play in each of the four animals when they are considered individually. In the original manuscript there was a typographical error in Table 2 where the p-value for animal R505 was listed as p=0.38 and not p=0.038 as it should have been. The proportions from the binomial tests *were* correctly entered in the table, but the p-value for this one animal was not. We are extremely sorry for the confusion caused and have very carefully re-checked the manuscript and supplementary information finding no further errors.

b) We acknowledge that one animal has a lower number of spiking events (44 events for the cued arm) than the others. However, this is comparable to the number of events per animal published in previous studies of replay, hence we do not disregard the data.

c) We have also re-analysed our data based on the reviewers’ suggestions to ensure that we have significant pre-play based on multiple measures. All the additional analyses support our main findings.

d) We have re-worded our interpretation of the results, such that we no longer relate them directly to *planning* of desired future paths leading to a goal. Instead we describe the preferential activation of place cell sequences, on an unvisited section of an environment, that are *associated* with a future goal. In light of this, we have updated the title of our revised manuscript to ‘Hippocampal Place Cells Construct Reward related Sequences through Unexplored Space’.

e) We have now included a discussion of the possible reasons for why we obtain a different result to the studies by Dragoi and Tonegawa (2011, Nature; 2013, PNAS; 2013, eLife).

*1) According to*
Table 2*, in one animal (R505) the number of sequences representing the cued locations are not significantly higher than chance. Moreover, in another animal (R1838) only 9 such sequences were detected so it seems that the recording was not optimal and this argues for excluding this latter animal from the dataset. Therefore, only 2 out of the 4 animals firmly support the conclusions of the work. Moreover, in the REST1 sessions in two animals these numbers were significant (one cued and one uncued arm). Therefore, more data are needed to convincingly demonstrate the conclusions of this work. At least showing it in one more animal to have n=3 showing the results could be sufficient*.

We would like to apologise again for the typographic error in the supplementary information. In Table 2, we stated the p-value of R505 for the cued arm in REST2 was 0.38, whereas it should actually have been 0.038. This is based on a binomial test, assessing whether the 38 significant preplay events out of 631 spiking events we obtained for this animal (6.02%) is significantly higher than the proportion expected by chance (4.57%). We greatly regret making this error and have comprehensively reviewed all supplementary information as well as the main manuscript, finding no further errors. As a visual aid we have now formatted all significant p-values in the supplementary information in bold. If the reviewers require further re-assurance we would like to highlight that the information given in all the supplementary tables is sufficient to enable p-values to be calculated directly. We also note that although we made a mistake in the supplementary table our assertion in the main text of the original manuscript that “significant effects were found when animals were analysed individually” was, and indeed still is, correct for all animals.

With respect to R1838, we do appreciate that this animal only makes a small contribution to the dataset However, the number of events recorded for this animal (44 for the cued arm) is similar to the number of events recorded for individual animals in previous studies in the field (e.g. [18] (n = 33); Dragoi and Tonegawa, 2011 (n = 34); Karlsson and Frank, 2009 (n = 22)). Additionally, despite the smaller n for this animal, we still see a significant bias of preplay for the cued arm vs the uncued arm during REST2. This is an effect that we see both at the population level, where we consider a total of 1628 events from four animals, and also for each of those animals individually. As such the preferential preplay of the cued arm is robust and extremely unlikely to have occurred by chance. Finally, even if R1838 were notionally discounted, which is not warranted, the remaining n=3 would be comparable to previous studies published in this field (e.g. Karlsson and Frank, 2009 (n = 3), Dragoi and Tonegawa, 2013, PNAS (n = 3), Diba and Buzsaki, 2007 (n = 3). In light of these points we have now updated the Results section of our manuscript to highlight that this animal provides fewer data than the other three.

The reviewers are correct that during REST1 (i.e. before the animals have any experience of the track) one animal exhibited significant preplay of the cued arm (R505) and one exhibited preplay of the uncued arm (R548); this information is shown in Table 2. These weak effects are only present in these individual animals and are not significant when the four animals are considered together, thus we did not report them beyond inclusion in the table. That said we note that this REST1 preplay is broadly consistent with the phenomenon first reported by Dragoi and Tongewa (2011, Nature) and subsequently reported by Dragoi and Tonegawa (2013, PNAS; 2013, eLife); where de novo preplay of place cell sequences for a novel and unseen environment was found, an effect believed to be driven by preconfigured hippocampal network states. We discuss de novo preplay below in more detail, in brief though: Because de novo preplay is thought to depend upon preconfigured cell assemblies as opposed to networks shaped by direct experience, we believe it to be a more subtle phenomena than replay of a familiar environment. Furthermore, the additional complexity of the environment used in this study, which included two visible but inaccessible arms, compared to those used in studies of de novo preplay, likely ameliorated the phenomenon further. Together these effects explain why we only see de novo preplay in these two animals. Additionally, because de novo preplay occurs before experiences of an environment it is unlikely to be biased by the motivational relevance of particular sections of that environment (e.g. a baited arm). Again this suggests why we saw de novo preplay for the cued as well as uncued arm. By contrast, the main effect we report—preferential preplay of the cued arm during REST2 (which occurs in all animals and only for the cued arm)—is significant when compared to REST1, and is driven by experience of the environment.

In sum, our dataset contains individually significant results from all (four) animals, exceeding the minimum number recommended by the reviewers (three). The weaker and unbiased de novo preplay observed in REST1 is consistent with the existing literature and could be expected; as pointed out by the reviewers. As such we do not believe further data collection is necessary. However, we have carefully addressed all the suggestions made by the reviewers regarding possible further analyses, details are below.

*The reviewers had the following recommendations to improve the robustness of the findings. One reviewer suggested that performing a sequence analysis animal-by-animal in the goal-cue run session would be more conclusive than it is in the REST2 session. In this case the claim could be related to prepay in the run session and this enough would be significant, not requiring the collection of more data*.

*Also, sequence analysis using Bayesian path prediction is more powerful than the currently used correlation analysis. Such analysis may provide more conclusive results for R505 as well in the REST2 session in which case further data may not be needed. This analysis should be pursued instead of or in addition to the correlation approach*.

This was an interesting and important suggestion. Previously we had believed that too few events were emitted during the short GOAL-CUE session to support a robust analysis. However, following the reviewers’ suggestion we analysed events recorded during GOAL-CUE while animals were at the end of the stem closest to the arms and were immobile or moving slowly (<10 cm/s). Although the number of events detected (172 events) was far fewer than during REST2 (1628) we still find significant preplay of the cued arm but not of the uncued arm: area under the curve (AUC) of the cumulative distribution of absolute correlations for spiking events for the cued arm is significantly smaller than that of the shuffled distribution (p = 0.02) but not for the uncued data (p = 0.24). Similarly a direct comparison of the proportion of preplay events emitted for the cued and uncued arm was close to significance (6.4% vs 4.12%, p = 0.052; see Table 4 for results for individual animals). These new analyses support and extend our existing results for the GOAL-CUE period, in which we showed an increased activity of cells with fields on the cued arm relative to cells with fields on the uncued arm (Figure 3). Thus, three different methods for analysing the GOAL-CUE period all indicate that preferential preplay of the motivationally relevant, cued arm was instantiated by the cueing. We agree with the reviewers that ‘in this case the claim could be related to preplay in the run session’ and have updated the manuscript and figures (Figure 3 and Figure 3—figure supplement 1) accordingly. Based on the clarifications and all the new analyses, we feel our claim that there is higher preplay of the cued arm is well justified. Therefore, we do not feel the collection of more data or further Bayesian analyses is needed to support our conclusions

*Another reviewer suggested that, given the small effect sizes, the authors should do additional analyses to validate their key results. There are two main analyses used to determine differences in preplay: first is a comparison of the proportion of significant preplay events. This method is sensitive to how the significance of each individual event is determined. The authors use rank correlation to compare the preplay event sequence to place cell templates in order to determine significance. The significance of each individual event should be determined by comparing to a distribution of correlations obtained by shuffling the cell order in the templates. Also, the significance value needs to be set at the p=0.025 level to control for multiple comparisons due to the two directions (Up and Down) of the run. How does this affect*
Figure 2*? Further, rat1838 has very few events, and presents as an outlier in this analysis*.

The reviewer describes the protocol we use to identify individually significant preplay events exactly. We would like to emphasise that to identify preplay events we use a two-tailed test (each tail being tested at the p < 0.025 level) to assess the significance of the correlation between the spike sequence and sequence of fields on the track. This test is appropriate because both forwards (positive correlation) and reverse (negative correlation) sequences are valid preplay events. Moreover, the reviewers highlight an important statistical concern, relating to the need to use a more stringent significance threshold if one conducts multiple comparisons on the same data (in this instance the reviewer is referring to the templates for UP and DOWN paths on the arms). However, we would like to clarify, the cells and the order in which they appear differs for the templates for the different running directions (i.e. the sequence of place cells for runs in the two directions are different and distinct, mean Spearman rank order correlation between UP and DOWN cued templates r = 0.05). Consequently, we tested across these with a set threshold of p < 0.025 for each tail. Finally, we would like to highlight the distribution-based analysis we use (AUC analysis) includes all spiking events and so also does not incur multiple comparisons in this situation.

*For the second distribution-based analysis, the authors do not directly compare the Cued and Uncued arm r-value distributions. Also, raw distributions should be shown (not just cumulative distributions), and can be compared using the Kolmogorov-Smirnov test*.

Thank you for pointing this omission out. We have now updated the Results section of the main text of our revised manuscript to include the statistics for comparing the cumulative distribution of absolute correlations for the cued and uncued (p = 0.02) arms in REST2. Finally, in the revised manuscript we have included figure supplements (Figure 2—figure supplement 1, Figure 3—figure supplement 1) showing raw correlation histograms for the distribution of correlations obtained for the cued arm against its shuffle distribution for the REST2 and GOAL-CUE periods, and carried out a Kolmogorov–Smirnov test as suggested (REST2 cued: p < 0.001; GOAL-CUE cued p = 0.031;)

*2) Another major concern speaks to evidence that the effect is functionally related to constructed representations of desired paths. Wording implying 'desired paths' may be an overstatement. The data do not speak directly to such a broad claim. It is also the case that there was as many forward and reverse sequence events detected. One might expect a preferential bias to forward event sequences if the function was representation of the unexplored path toward the reward. A strong recommendation would be to downplay the statements about constructed desired paths and also discuss the implication of reverse events being found in similar frequency, including alternative accounts like proposed by Foster and Wilson (2006)*.

We thank the reviewers for raising this point. We have revised the wording used to describe preplay of the cued arm and now simply state that place cell sequences associated with goals are preferentially preplayed; we refrain from relating this result directly to future planning and have amended the Title of the manuscript accordingly. Moreover, we have included a discussion on the finding of equal amounts of forward and reverse events (as well as event for UP and DOWN paths), and discussed its relationship with theoretical accounts proposed by Foster and Wilson (2006) and Foster and Knierim (2012) where reverse replay is proposed as a mechanism for reinforcement learning and forward replay as planning for future behaviour.

*3) The results contradict previous results on preplay, and this is not addressed in the manuscript. The conclusion that visual observation and goal cueing is necessary for observing preplay of a future experience contradicts previous results from Dragoi and Tonegawa (2011, Nature; 2013, PNAS; 2013, eLife) that show that preplay of a novel experience occurs without these conditions. While the lack of preplay of the uncued arm in the current study can be attributed to biasing of preplay towards a more motivationally relevant arm, the lack of preplay of the STEM arm during REST1 (*Figure 2—figure supplement 2*; and*
Table 3*) directly contradicts previous results. The previous interpretation was that pre-existing temporal firing sequences in the hippocampus (cellular assemblies based on functional connectivity) become bound to novel environments as place cell sequences. But the current results force a re-interpretation that preplay of a novel environment only occurs if it is visually observed. This needs to be explicitly addressed in the manuscript, and requires a careful re-examination of the analysis*.

We thank the reviewers for suggesting a discussion on this apparent contradiction with previous studies. We would like to emphasise that our results do not speak against the hypothesis that pre-existing temporal firing sequences exist in the hippocampus, as suggested by other authors (e.g. Dragoi and Tonegawa, 2011, Nature). In our manuscript we describe preferential preplay of a motivational relevant but unexplored section of the environment, an effect that is contingent on the animal having seen the environment. However, we do see marginally significant preplay of the stem during REST1, an effect consistent with de novo preplay (area under the curve analysis of stem preplay vs shuffle in REST1: p = 0.053). We have now added a section to the Discussion where we set out the likely reasons for the difference between our results and those of Dragoi and Tonegawa (2011, Nature; 2013, PNAS; 2013, eLife). Briefly, we believe that de novo preplay is a relatively subtle phenomenon, having gone undetected in a number of previous studies (e.g. Wilson and Mcnaughton, 1994; Lee and Wilson, 2002).

Furthermore, the complexity of the environment we used (an accessible stem, with visible arms blocked by a barrier) may have ameliorated the effect further. However despite this we do see marginally significant preplay of the stem during REST1.

[Editors’ note: the author responses to the re-review follow.]

*Thank you for resubmitting your work entitled* “*The Hippocampus Constructs Desired Paths through Unexplored Space*” *for further consideration at eLife. Your revised article has been evaluated by Eve Marder (Senior editor) and a member of the Board of Reviewing Editors, and two peer reviewers. The manuscript has been improved but there are some remaining issues that need to be addressed before acceptance, as outlined below: Briefly, both of the peer reviewers felt that the recommended Baysian analysis is a requisite*.

*The authors have filled some of the missing gaps in the analysis based on the reviewers’ comments, and the manuscript is substantially improved. However, the concerns about the small effect sizes and the robustness of the results still remain. This is also apparent in the raw distributions that the authors now show in*
Figure 2—figure supplement 1
*and*
Figure 3—figure supplement 1*. A more-powerful Bayesian based decoding analysis (originally the suggestion of Reviewer 1) is necessary to confirm these results, and this will have a substantial impact on how these results will be received*.

We appreciate the reviewers’ positive comments and suggestion to use a Bayesian approach for the decoding analysis. We have now done exactly this and are happy to report that Bayesian decoding of the preplay events finds the same bias towards the cued arm as the rank-order method we previously used (full details and statistics are provided below). Specifically, during REST2 the cued arm is significantly preplayed while the uncued arm is not. Importantly, preplay of the cued arm only develops after the animal witnesses the arm being baited and is not present during REST1.

*The correlation based template matching method that the authors have used in the manuscript is less effective than Bayesian decoding in detecting sequential activation/ reactivation (e.g. Carr et al., Nat. Neurosci. 2011, Vol. 14, pp 147–153). There is a concern that the rank correlations-based method may lead to false positive results due to other factor such as the burst propensity differences between cells. The fact that significant preplay events were seen in REST1 in one cued and another uncued arm is still hard to interpret. This may indeed show preplay as in the Dragoi studies but then why don't we see it in all animals? And why don't we see significant preplay of the uncued arms in REST2 too? Of course there is the other interpretation that significant preplay in REST1 may represent false positive detection, which may question how consistently one may see the preferential preplay of the cued arm (in REST2) if the experiments are repeated many times*.

Again we thank the reviewers for their comments. Clearly Bayesian decoding complements the rank-order method already reported in the paper: the rank-order method uses only the first spike emitted by each cell in conjunction with the field peaks, as such it makes few assumptions about phenomena such as bursting. In contrast the Bayesian approach uses all spikes and the full extent of the firing ratemap, for this reason it is plausibly more powerful but does implicitly include assumptions. For example, that the bursting dynamics seen in online running will be present in preceding periods of offline preplay. That said, when applied to our data, both approaches reveal the same main effects: in REST2 the cued arm is exclusively preplayed, an effect that is not present in REST1. Furthermore, under the Bayesian analysis, none of the animals individually exhibited significant preplay for either the cued or uncued arm during REST1. Similarly, none exhibited preplay of the uncued arm in REST2. These additional details are completely consistent with the interpretation we had previously placed on our data: that we did not see preplay as reported by Dragoi et al in REST1 because of the complexity of our apparatus and that in REST2 preplay was ‘captured’ by the cued arm such that preplay of the uncued arm was not seen.

*Bayesian decoding uses all of the spikes that occur during a reply event, and the entire place field structure to estimate the reactivated sequence of positions. In contrast, the correlation method compare the sequence of first spike of each cell during replay, and the relative location of place field centers, and is therefore less effective when only a subset of neurons participate in the replay event. Although the authors mention that they use multiple methods to compare the preplay of the cued arm to the uncued arm, the core method still remains the same, the correlation based template matching. Based on the output of this correlation-based analysis, the authors then evaluate the results using either the comparison of the number of significant events, or the distribution of the r-values. Comparing the distributions is a more powerful evaluation, and the differences between the cued and uncued arm preplays are small. Moreover, the difference in data and shuffle distributions are also small for Cued Arm preplay in Rest 2 (*Figure 2—figure supplement 1*). As a comparison, for previous results in* de novo *preplay, (Dragoi and Tonegawa, 2011, Nature), which the authors mention might be a relatively more subtle phenomenon, the preplay effect was stronger based on the distribution based analysis*.

*The authors have sufficient data to run a Bayesian based decoding analysis in at least 3 of the animals. Therefore, my strong recommendation would be to evaluate their data using this more powerful approach. Confirmation of their results using this method will substantially alleviate any concerns regarding the robustness of this effect, and make a stronger impact*.

We thank the reviewers for their clear and constructive comments. As stated above, we agree that a Bayesian decoding approach is a powerful tool to complement our existing rank-order analyses. However, as highlighted by this reviewer, it does require a larger number of place cells to be held in parallel in order to generate robust results. Accordingly, as suggested, we applied Bayesian decoding to the spiking events recorded during REST1 and REST2 for the 3 animals with the highest cell yield (R1838 excluded). First, to confirm the validity of the approach and its correct implementation, we applied the same Bayesian framework to online track running (RUN2). The algorithm accurately estimated position (mean cued arm error = 10.0 cm, mean uncued arm error = 10.0 cm, for 1 cm spatial bins, Figure 4). Subsequently, to score preplay events, we applied a line fitting algorithm and flagged preplay events as those exceeding the 95^th^ percentile of their own null distribution generated by shuffling the cell identities (following Davidson et al. 2009, Figure 4). Results from the Bayesian approach were as follows. In REST2 we found significantly more preplay events for the cued arm compared to the uncued arm (Cued = 7.64%, p < 0.001, Uncued = 4.78%, p < 0.55, Cued vs. Uncued, p < 0.001). Conversely, during REST1 neither arm exhibit preplay (Cued = 5.04%, p = 0.43, Uncued = 4.69%, p = 0.55, Cued vs. Uncued, p = 0.30, Figure 4). Finally the cued arm exhibited significantly elevated preplay in REST2 vs. REST1 (p < 0.001). These results exactly match those obtained with the rank-order analysis described in the earlier version of our manuscript, strongly reinforcing our previous conclusions that during rest the hippocampus preplays reward-related sequences.